# WORLDCRAFTER: DYNAMIC SCENE GENERATION FROM A SINGLE IMAGE WITH GEOMETRIC AND TEMPORAL CONSISTENCY

## ABSTRACT

We present WorldCrafter, a novel framework that enables interactive dynamic scene generation from a single image by leveraging geometry-aware and temporal modeling. Existing methods often suffer from texture distortion, structural inaccuracies, and temporal flickering under large viewpoint changes. These issues mainly caused by explicit pixel-wise reprojection strategies. To address these challenges, WorldCrafter introduces two complementary modules: 1) *Geometry-aware Video Depth Refinement*, which enhances structural fidelity by refining depth with multi-frame geometric priors and semantic cues; and 2) *Object-consistent Temporal Modeling*, which disentangles video frames into object-level layers to improve coherence between static backgrounds and dynamic foregrounds. These components form a unified rendering-inpainting framework for photorealistic and camera-controllable dynamic scene generation. Experiments demonstrate that WorldCrafter produces geometrically accurate and temporally coherent results across diverse scenes and camera trajectories.

## 1 INTRODUCTION

Recent progress in world models (Yu et al., 2025; Shriram et al., 2025; Yu et al., 2024a; Chung et al., 2023; Yu et al., 2023; Höllein et al., 2023; Liang et al., 2025) has enabled the generation of photorealistic and camera-controllable scenes by leveraging depth estimation, segmentation, and inpainting. However, most of these methods focus on *static* scenes and lack the ability to model temporal dynamics, making them unsuitable for scenarios involving motion or large viewpoint changes. This gap arises from the inherent challenges of modeling dynamic geometry and ensuring spatial-temporal coherence across frames. To bridge this gap, we take a step forward in *dynamic* scene generation by unifying geometry-aware modeling with spatial-temporal consistency.

Meanwhile, video world models (Xiao et al., 2025; YU et al., 2025; Yu et al., 2024b; Bai et al., 2025; Mao et al., 2025; Han et al., 2025)

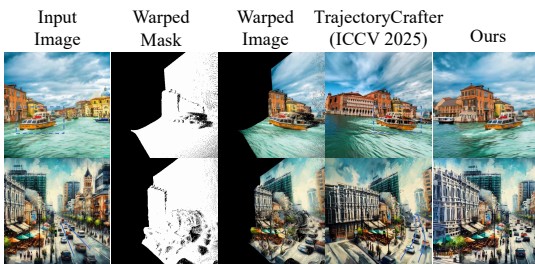

Input Image  Warped Mask  Warped Image  TrajectoryCrafter (ICCV 2025)  Ours

Figure 1: **Motivation.** Existing pixel-wise reprojection methods (e.g., TrajectoryCrafter (YU et al., 2025), TrajectoryAttention (Xiao et al., 2025)) often struggle with geometric distortions and temporal inconsistency. As shown, a $60°$ pan-left introduces noticeable artifacts, including boat deformation (top) and a curved road (bottom). These failures reveal the limitations of explicit reprojection and motivate the pursuit of geometry-aware and temporally coherent generation. Our method alleviates these issues, with consistent results (right).

move beyond static scene synthesis, generating dynamic scenes through warping-based pipelines guided by depth and camera pose estimations (e.g., DUSt3R (Wang et al., 2024), MASt3R (Leroy et al., 2024)). These approaches compute transformation maps from camera pose changes and point maps, but their direct pixel warping often results in flickering, distortion, and structural collapse under large viewpoint shifts (See Figure 1, 2).

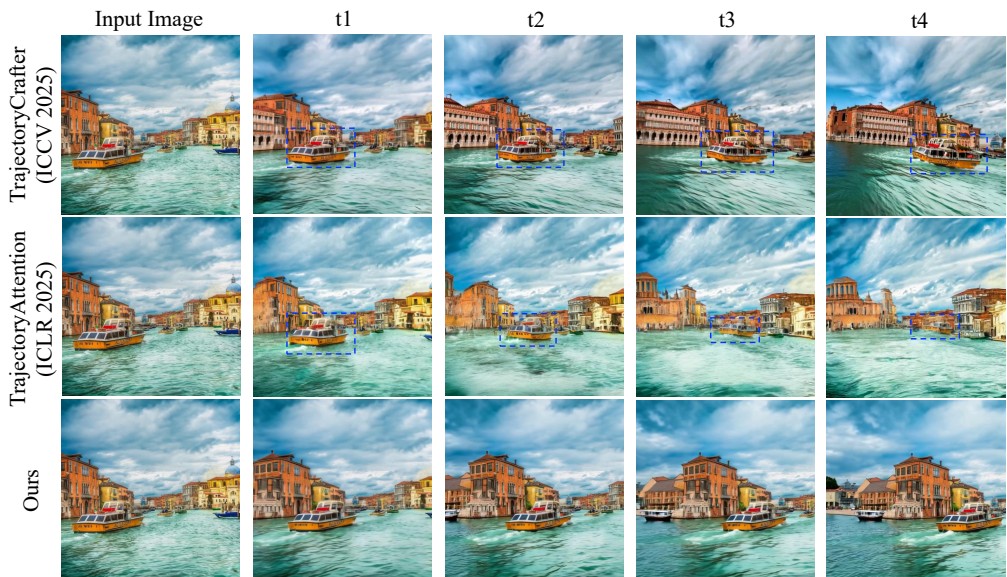

Prompt: *The Grand Canal in Venice, boats, historical buildings, waterway.*

Figure 2: **Comparison with camera-controlled video generation methods.** Existing methods often produce distortions without geometry-aware constraints and spatial-temporal guidance. Our WorldCrafter interactively generates dynamic scenes with geometric and temporal consistency, aligned with the input image, text prompt and camera trajectory.

In this paper, we propose WorldCrafter, an interactive framework for dynamic scene generation from a single image with user-defined camera trajectories. Unlike previous methods (Xiao et al., 2025; YU et al., 2025) that rely on explicit pixel-wise reprojection, our WorldCrafter employs geometry-aware depth refinement and object-consistent temporal modeling with a video diffusion framework. This unified design enables temporally coherent and geometrically consistent generation even under large camera motion. To ensure both geometric and temporal consistency, we introduce two key components: 1) A *geometry-aware video depth refinement* module that improves spatial coherence within semantic regions and stabilizes depth across frames. 2) An *object-consistent temporal modeling* module that uses geometric priors, segmentation constraints, and temporal cues to guide inpainting. This enables object-level consistency and coherent novel-view video synthesis.

In particular, the geometry-aware video depth refinement module integrates semantic cues into the generative process in a soft and implicit manner. Instead of explicit pixel-wise warping, this module produces temporally aligned depth features that guide the video diffusion model towards consistent geometry-aware scene synthesis over time. The object-consistent temporal modeling module explicitly decomposes the scene into static backgrounds and dynamic foregrounds. By disentangling object motion from scene structure, this module improves temporal depth refinement and enhances consistency across objects. This design enables fine-grained control of object dynamics, which is crucial for generating coherent scene videos under complex camera trajectories.

Our contributions are as follows:

- We introduce WorldCrafter, an interactive framework for dynamic scene generation from a single image with geometric and temporal consistency.

- We propose a geometry-aware video depth refinement module that achieves temporally stable depth without relying on explicit pixel-wise reprojection.

- We introduce object-consistent temporal modeling that separates foregrounds from backgrounds, enhancing object-to-object coherence and controllability.

- Extensive experiments demonstrate that WorldCrafter produces consistent dynamic scenes from a single image, preserving geometry- and object-consistent results under large camera motions and diverse scenarios.

## 2 RELATED WORKS

**Static 3D World Generation.** Interactive 3D world models aim to synthesize realistic environments for user exploration. Recent works have primarily focused on generating *static* 3D scenes from images or text prompts. WonderWorld (Yu et al., 2025) introduced depth-conditioned diffusion for scene synthesis, while WonderJourney (Yu et al., 2024a) adopted point-based rendering to enhance view consistency. LucidDreamer (Chung et al., 2023) and RealmDreamer (Shriram et al., 2025) leveraged domain-agnostic point clouds and depth priors for generalizable scene generation. Text2Room (Höllein et al., 2023) and Text2NeRF (Zhang et al., 2024) employed modular pipelines to create room-scale environments from text, and DreamScene (Li et al., 2024) introduced a text-to-3D framework with formation pattern sampling. PhotoconsistentNVS (Yu et al., 2023) improved multi-view alignment through autoregressive diffusion, while Wonderland (Liang et al., 2025) applied 3DGS for efficient reconstruction. Although effective for static settings, these methods lack temporal modeling and cannot handle dynamic content or camera motion.

**Video World Models.** Extending static world models to dynamic scene generation introduces challenges such as occlusion, geometric distortion, and temporal flickering. Video world models are typically formulated as video prediction tasks, synthesizing future frames conditioned on camera trajectories or text prompts. Trajectory-Attention (Xiao et al., 2025) and TrajectoryCrafter (YU et al., 2025) employed attention mechanisms and dual-stream diffusion to improve camera control, but both rely on pixel-wise reprojection, which often fails under fast motion or occlusion. ViewCrafter (Yu et al., 2024b) and ReCamMaster (Bai et al., 2025) enhanced realism through trajectory supervision and diffusion priors, yet depend heavily on synthetic multi-view datasets. Free4D (Liu et al., 2025b) proposed a tuning-free, point-guided video diffusion pipeline but struggled with structural realism. VideoScene (Wang et al., 2025b) distilled spatial-temporal structure into video outputs, while GCD (Van Hoorick et al., 2024) enforced consistency with geometric priors. NWM (Bar et al., 2025) focused on egocentric view prediction for downstream planning tasks. Overall, most video world models lack geometry-guided modeling, leading to temporal inconsistency and structural artifacts. In contrast, our approach incorporates geometry-aware depth refinement to enable coherent dynamic scene generation.

## 3 PROPOSED METHODOLOGY

### 3.1 PRELIMINARIES

**Video diffusion models** (Song et al., 2020; Blattmann et al., 2023; Hong et al., 2022; Yang et al., 2024) generate temporally coherent videos by denoising a sequence of latent representations through a stochastic reverse process. Formally, given a video $x_0 \in \mathbb{R}^{L \times 3 \times H \times W}$, the forward process gradually adds Gaussian noise to $x_0$, resulting in a noisy sample $x_t$ at timestep $t$:

$$x_t = \sqrt{\bar{\alpha}_t} x_0 + \sqrt{1 - \bar{\alpha}_t} \epsilon, \quad \epsilon \sim \mathcal{N}(\mathbf{0}, \boldsymbol{I}), \tag{1}$$

where $\bar{\alpha}_t$ is a cumulative product of the noise schedule coefficients and $t \in [0, 1]$ denotes the continuous diffusion time step. The reverse process is learned through a neural network $\epsilon_\theta(x_t, t)$ trained to predict added noise, with the denoising objective:

$$\mathcal{L}_{\text{denoising}} = \mathbb{E}_{t, \epsilon} \left[ \| \epsilon_\theta(x_t, t) - \epsilon \|_2^2 \right]. \tag{2}$$

We implement $\epsilon_\theta$ using the diffusion Transformer DiT (Peebles & Xie, 2023) that is originally developed for high-resolution image synthesis. Its self-attention mechanism effectively models long-range spatial and temporal dependencies making it well-suited for interactive dynamic scene generation. Building on this, we employ a pre-trained latent video diffusion model to generate temporally evolving scenes from a single image. We adapt the model to support interactive dynamic scene generation with user-specified camera trajectories and text prompts.

### 3.2 OUR WORLDCRAFTER

**Overview.** Figure 3 illustrates our WorldCrafter, an interactive framework that generates dynamic scenes from a single image with user-specified camera trajectories and text prompt. The framework

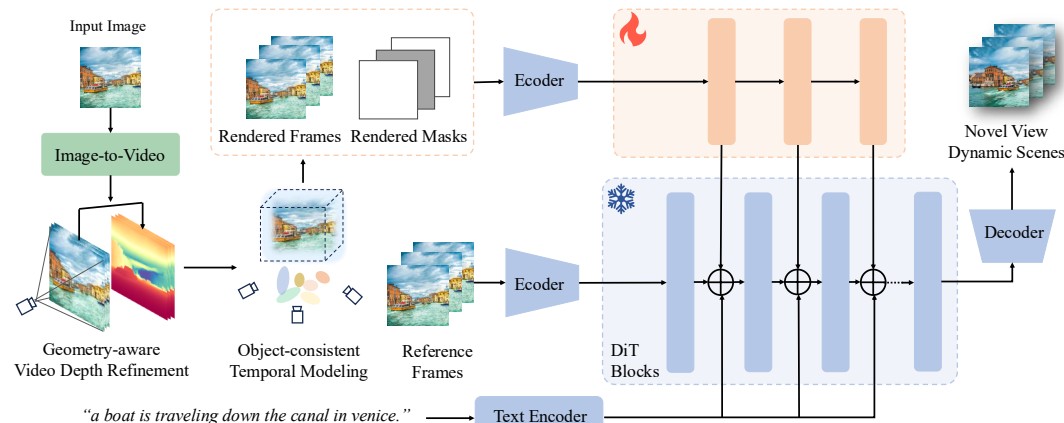

Figure 3: **Overview of the proposed WorldCrafter.** From a single input image, an Image-to-Video model generates an initial video whose outputs serve as reference frames. WorldCrafter introduces two key modules: (1) geometry-aware video depth refinement, which estimates and refines depth maps to preserve geometric structure, and (2) object-consistent temporal modeling, which leverages object masks and reference frames to enforce spatial-temporal coherence. These modules jointly enable controllable and photorealistic novel-view scene generation.

integrates a geometry-aware depth refinement module that enforces geometric consistency with an object-consistent temporal modeling strategy that disentangles dynamic motions from static backgrounds, enhancing controllability and spatio-temporal coherence. These components jointly enable geometry-aware, temporally consistent dynamic scene generation.

**Motivation.** Despite recent progress in 3D world models (Yu et al., 2025; Shriram et al., 2025; Yu et al., 2024a; Chung et al., 2023; Yu et al., 2023; Höllein et al., 2023; Liang et al., 2025; Liu et al., 2025a), most approaches remain limited to *static* scenes and struggle to capture temporal dynamics or handle large camera motions robustly. Pixel-wise reprojection methods (YU et al., 2025; Xiao et al., 2025), which estimate optical flow from changes in depth and pose of the camera, often lead to geometric distortions and temporal flickering (see Figure 1).

To overcome these limitations, we propose a unified framework for dynamic scene generation with three objectives: (1) controllability from a single image, (2) temporal and geometric consistency, and (3) interactive control through decomposed scene components. In contrast to pixel-wise reprojection methods such as TrajectoryCrafter (YU et al., 2025) and TrajectoryAttention (Xiao et al., 2025), which suffer from deformation and bending artifacts under large viewpoint changes, our approach refines depth and enforces object consistency through 3D geometry-aware and object-level modeling. By avoiding explicit pixel-wise reprojection, our framework achieves coherent, geometry-aware, and temporally consistent dynamic scene generation from a single image.

**Formulation.** We formulate interactive dynamic scene generation as a sequential process, where the model predicts the next frame $\mathbf{V}_{t+1}$ conditioned on the current observation and user input. Unlike WonderWorld (Yu et al., 2025) which focuses on *static* 3D world synthesis, our formulation explicitly models temporal *dynamics*, capturing how the scene evolves over time. At each timestep $t$, the model receives a triplet $(\mathcal{C}_t, \mathcal{N}_t, \mathcal{L}_t)$, where $\mathcal{C}_t = \{\mathbf{I}_t, \mathcal{P}_t\}$ denotes the current scene, consisting of the image $\mathbf{I}_t$ and a scene-level prompt $\mathcal{P}_t$. $\mathcal{N}_t$ specifies the desired next-scene content, such as object actions or scene transitions. $\mathcal{L}_t = \{\mathcal{T}_t, \mathcal{Y}_t\}$ provides layout guidance, where $\mathcal{T}_t = (\mathbf{C}_1, \ldots, \mathbf{C}_N)$ is a sequence of camera poses and $\mathcal{Y}_t$ is a text description of the desired camera trajectory. The next scene is then generated by the world model:

$$\mathbf{V}_{t+1} = \mathbf{G}_{\text{world}}(\mathbf{W}_{\text{update}}(\mathcal{C}_t, \mathcal{N}_t, \mathcal{L}_t)), \quad (3)$$

where $\mathbf{W}_{\text{update}}$ is the scene state updater (e.g., latent encoding or motion modeling), and $\mathbf{G}_{\text{world}}$ is the rendering module that synthesizes the next dynamic scene.

### 3.3 GEOMETRY-AWARE VIDEO DEPTH REFINEMENT

Given a video $V = \{I_t\}_{t=1}^T$ generated by an Image-to-Video model (Wang et al., 2025a; Team., 2025), our goal is to lift it into a dynamic scene representation by refining its depth sequence in

Figure 4: **Geometry-aware depth refinement and object-consistent temporal modeling**. A depth estimator and a segmentation network generate depth maps and object masks, which are refined by region grouping and median filtering. The refined cues are then integrated through object-consistent temporal modeling to render spatially coherent and temporally stable frames.

a way that preserves both geometric accuracy and temporal consistency. To this end, we adopt DepthCrafter (Hu et al., 2025) as our base video depth estimator, which predicts an initial depth map $D_t$ for each frame $I_t$. Although DepthCrafter provides coherent depth predictions, it often introduces geometric artifacts, including flying points, depth leakage across object boundaries, and structural distortions under large viewpoint changes.

To address these issues, we propose a geometry-aware video depth refinement module guided by semantic segmentation cues (See Figure 4). For each frame $I_t$, we apply SAM (Kirillov et al., 2023) to obtain a set of binary masks $\{\mathcal{S}_j\}_{j=1}^{N_s}$, where each $\mathcal{S}_j \in \{0,1\}^{H \times W}$ corresponds to a distinct semantic region. These masks group pixels with similar geometric properties, enabling more consistent depth refinement within each segment. The key insight is that semantic regions in video frames, such as walls, vehicles, or trees, often correspond to physically coherent structures that share similar depth distributions. To leverage this prior, we enforce intra-segment depth consistency to reduce artifacts and enhance geometric alignment across frames. We refine the depth for each object-level region using a filtering operator such as Median Filtering (Huang et al., 1979).

Given the initial depth map $D_t$ at time $t$ and a set of $N_s$ semantic masks $\{\mathcal{S}_j\}_{j=1}^{N_s}$ predicted by SAM, the refined depth $\hat{D}_t$ is computed as:

$$\hat{D}_t(x,y) = \sum_{j=1}^{N_s} \mathcal{S}_j(x,y)\, \mathcal{R}(D_t, \mathcal{S}_j)(x,y)$$

$$+ \left(1 - \sum_{j=1}^{N_s} \mathcal{S}_j(x,y)\right) D_t(x,y), \tag{4}$$

where $\sum_{j=1}^{N_s} \mathcal{S}_j(x,y)$ indicates whether pixel $(x,y)$ is covered by any semantic mask. Since the masks are binary and non-overlapping, the sum evaluates to either 0 or 1, thus selecting the refined or original depth accordingly. The operator $\mathcal{R}(D_t, \mathcal{S}_j)(x,y)$ denotes the refined depth of pixel $(x,y)$ within region $\mathcal{S}_j$, typically defined as

$$\mathcal{R}(D_t, \mathcal{S}_j)(x,y) = \mathrm{median}\big\{D_t(x',y') \mid \mathcal{S}_j(x',y') = 1\big\}.$$

This refinement promotes spatial coherence within semantic regions and enhances temporal stability across video frames. It preserves geometric boundaries, suppresses floating artifacts, and exploits segment-level structural priors for more consistent depth estimation. Ablation studies further demonstrate that the refined depth maps supply explicit geometric cues that benefit both geometry-aware rendering and dynamic scene generation.

### 3.4 Object-consistent Temporal Modeling

We leverage geometric, semantic, and temporal cues to guide inpainting, ensuring object-level consistency and coherent novel-view synthesis. Given video frames $\{I_t\}_{t=1}^T$, their refined depth maps $\{\hat{D}_t\}_{t=1}^T$, and estimated camera parameters $\{C_t = (K_t, R_t, T_t)\}_{t=1}^T$ with intrinsic matrix $K_t$ and extrinsics $(R_t, T_t)$, we incorporate per-frame semantic masks $\{\mathcal{S}_j^t\}_{j=1}^{N_s}$ from SAM (Kirillov et al.,

2023; Lan et al., 2024). For each frame $t$, pixels $(x, y)$ are back-projected into 3D and assigned to their corresponding semantic segment:

$$\mathbf{X}_t^j(x, y) = R_t^{-1} \left( \hat{D}_t(x, y) \cdot K_t^{-1} \begin{bmatrix} x \\ y \\ 1 \end{bmatrix} - T_t \right), \quad \text{if } \mathcal{S}_j^t(x, y) = 1. \tag{5}$$

These 3D points are grouped and encoded as segment-conditioned 3D Gaussians:

$$\mathcal{G}_t^j = \left\{ \left( \mu_i^j, \Sigma_i^j, \mathbf{c}_i^j, \alpha_i^j \right) \mid (x, y) \in \text{supp}(\mathcal{S}_j^t) \right\},$$

where each Gaussian is defined by position $\mu_i^j$, covariance $\Sigma_i^j$, color $\mathbf{c}_i^j$, and opacity $\alpha_i^j$ within segment $j$. Here, $\text{supp}(\mathcal{S}_j^t)$ denotes the support of binary mask $\mathcal{S}_j^t$, i.e., pixels with $\mathcal{S}_j^t(x, y) = 1$. To render a novel view at pose $C_t = (K_t, R_t, T_t)$, we aggregate Gaussians from previous frames and segments via the 3DGS renderer (Kerbl et al., 2023; Yu et al., 2025):

$$\tilde{I}_t^{\text{view}}(x, y) = \mathcal{R}_{\text{3DGS}} \left( \bigcup_{t=1}^{\tau-1} \bigcup_{j=1}^{N_s} \mathcal{G}_t^j, \; C_t \right),$$

where $\mathcal{G}_t^j$ denotes the Gaussian representation of pixel $(x, y)$ at time $t$. The renderer integrates visibility, depth, and appearance to produce object-aware novel views with temporal consistency. This design enables object-consistent reconstruction by combining geometric alignment with learned priors in the 3D Gaussian representation.

Due to occlusions and visibility constraints, $\tilde{I}_t^{\text{view}}$ may contain missing or uncertain regions. To address this, we compute a binary mask $M_t \in \{0, 1\}^{H \times W}$, where each pixel indicates the absence of confident rendered content. Unlike traditional warping-based masks, $M_t$ is derived directly from undistorted 3D object geometry, enabling more accurate detection of occluded regions. We then apply a rendering-guided video inpainting module based on a modified CogVideoX (Yang et al., 2024), which takes the rendered frame $\tilde{I}_t^{\text{view}}$, visibility mask $M_t$, and a text prompt $p$ as inputs:

$$\hat{I}_t = \mathcal{I}_{\text{net}}(\tilde{I}_t^{\text{view}}, M_t, p),$$

and outputs the final inpainted frame $\hat{I}_t$. This module extends CogVideoX to our geometry-aware pipeline, enabling high-fidelity synthesis with temporal coherence and fine-grained object consistency. As shown in Table 3, this integration improves Subject Consistency and Imaging Quality on VBench by bridging geometry-aware refinement with object-consistent modeling.

## 4 EXPERIMENTS

**Evaluation Metrics and Dataset.** We evaluate all models on the WonderWorld (Yu et al., 2025) dataset using VBench (Huang et al., 2024) and CLIP (Radford et al., 2021) to assess temporal coherence, view consistency, and text alignment. Specifically, CLIP-T measures frame-to-text alignment, CLIP-F computes temporal coherence via similarity between adjacent frames, and CLIP-V evaluates consistency between the source and generated views.

**Implementation Details.** We use single images and prompts from the WonderWorld (Yu et al., 2025) dataset and follow official inference pipelines to simulate six camera motions pan (left/right/up/down) and zoom (in/out) for fair comparison. Our framework is built on WonderWorld, using CogVideoX-Fun-5B (Yang et al., 2024) as the video diffusion model, DepthCrafter (Hu et al., 2024) for video depth estimator, and used SAM (Kirillov et al., 2023) as the segmentation network. All experiments were conducted on an NVIDIA RTX 6000 GPU (48 GB). Additional details are provided in the supplementary material.

### 4.1 QUANTITATIVE COMPARISONS

We evaluate WorldCrafter against both interactive 3D world models and camera-controlled video generation methods, focusing on semantic alignment, temporal coherence, and structural consistency. As shown in Table 1, WorldCrafter consistently outperforms state-of-the-art baselines across

Table 1: CLIP-based quantitative comparison on WonderWorld (Yu et al., 2025). "CLIP-T": text alignment, "CLIP-F": temporal coherence, "CLIP-V": view consistency. Higher is better. Best in **bold**, second-best underlined.

| Method | CLIP-T ↑ | CLIP-F ↑ | CLIP-V ↑ |
|---|---|---|---|
| TrajectoryCrafter (YU et al., 2025) | 28.01 | 96.83 | 86.60 |
| ReCamMaster (Bai et al., 2025) | 30.54 | 99.14 | 93.35 |
| TrajectoryAttention (Xiao et al., 2025) | 30.24 | 98.91 | 91.37 |
| WonderJourney (Yu et al., 2024a) | 31.08 | 96.09 | 94.11 |
| WonderWorld (Yu et al., 2025) | 30.16 | 97.40 | 93.88 |
| WorldCrafter (Ours) | **31.49** | **99.34** | **95.81** |

Table 2: **Quantitative comparison using VBench (Huang et al., 2024).** We evaluate on the in-the-wild image benchmark from the WonderWorld (Yu et al., 2025) dataset and report VBench (Huang et al., 2024) scores for results generated along novel trajectories. "Camera." denotes camera-controlled video generation, while "3D." refers to interactive 3D scene generation. The best results are shown in **bold**, and the second-best are underlined.

| | Method | Subject Consistency ↑ | Background Consistency ↑ | Motion Smoothness ↑ | Imaging Quality ↑ |
|---|---|---|---|---|---|
| Camera. | TrajectoryCrafter (YU et al., 2025) (ICCV 2025) | 0.8967 | 0.9413 | 0.9841 | 0.7323 |
| | ReCamMaster (Bai et al., 2025) (ICCV 2025) | 0.9017 | 0.9457 | **0.9903** | 0.6902 |
| | TrajectoryAttention (Xiao et al., 2025) (ICLR 2025) | 0.8744 | 0.9127 | 0.9662 | 0.7022 |
| 3D. | WonderJourney (Yu et al., 2024a) (CVPR 2024) | 0.9108 | 0.9402 | 0.9380 | 0.7062 |
| | WonderWorld (Yu et al., 2025) (CVPR 2025) | 0.9330 | 0.9488 | 0.9687 | 0.7094 |
| | WorldCrafter (Ours) | **0.9463** | **0.9685** | 0.9885 | **0.7466** |

all CLIP-based metrics, where higher values reflect stronger text alignment and visual coherence. Notably, CLIP-V improves from 93.88 in WonderWorld to 95.81 with our approach, indicating superior view consistency under novel camera trajectories. To further assess interactive dynamic scene generation, we benchmark using VBench (Huang et al., 2024), which evaluates subject consistency, background consistency, motion smoothness, aesthetic quality, and imaging quality. Results in Table 2 show that our method achieves the highest performance in subject consistency, background consistency, and overall perceptual quality. For example, background consistency improves from 0.9488 in WonderWorld to 0.9685 in WorldCrafter, while imaging quality increases from 0.7094 to 0.7466. These gains highlight the effectiveness of our geometry-aware and temporal modeling strategies in reducing artifacts such as incoherent structures and object duplication. Overall, the results show that *WorldCrafter* surpasses existing methods and delivers high-fidelity and temporally consistent dynamic scenes, establishing a new framework for interactive video world modeling.

## 4.2 QUALITATIVE COMPARISONS

We qualitatively compare WorldCrafter with interactive 3D world models and camera-based video generation methods under large viewpoint changes (Figures 2, 5, and 6). As shown in Figure 2, baseline methods often produce unstable or distorted frames due to limited geometry constraints and weak spatial-temporal modeling. In contrast, WorldCrafter generates stable and spatially consistent sequences that remain faithful to the input image and prompt. Figure 5 highlights object-level inconsistencies in prior methods, where objects deform or change appearance across frames (blue dashed boxes). Our method maintains consistent object representations with strong spatial-temporal coherence. Figure 6 further illustrates the advantage of geometry-aware guidance. Camera-based methods (YU et al., 2025; Bai et al., 2025; Xiao et al., 2025) frequently suffer from artifacts under large viewpoint changes due to inaccurate warping, whereas our approach preserves geometric consistency and scene fidelity. Interactive 3D world models (Yu et al., 2024a; 2025) may produce high-quality single frames but lack temporal modeling, leading to inconsistent sequences. Static 3D models cannot produce dynamic scenes at all. Camera-based approaches rely on warping with estimated poses and depth, which often fail under complex motion. Our WorldCrafter achieves geometry-aware, object-consistent, and temporally coherent dynamic scene generation.

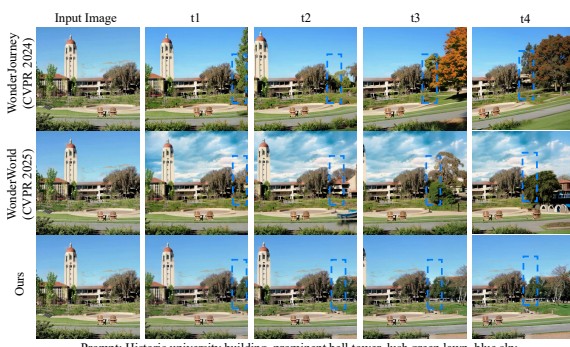

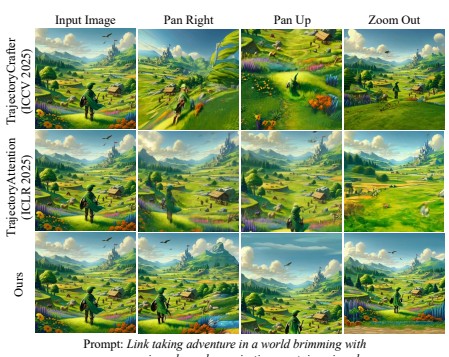

Figure 5: **Comparison with static 3D world models.** Baselines show object changes across time (dashed boxes), while ours preserve temporal coherence.

Figure 6: **Comparison with video world models.** Baselines suffer from artifacts under large viewpoint changes.

Table 3: **Ablation study.** "Subj.": Subject Consistency, "Back.": Background Consistency, "Smooth.": Motion Smoothness, "Img.": Imaging Quality, "Geom.": Geometry-aware, "Obj.-Consist.": Object-consistent, "Re.-Inpaint": Rendering-guided Video Inpainting.

| Method | Geom. | Re.-Inpaint | Obj.-Consist. | CLIP-T ↑ | CLIP-F ↑ | CLIP-V ↑ | Subj.↑ | Back.↑ | Smooth.↑ | Img.↑ |
|---|---|---|---|---|---|---|---|---|---|---|
| Baseline | | | | 31.20 | 96.51 | 94.80 | 0.9175 | 0.9452 | 0.9530 | 0.7072 |
| + Geometry-aware | ✓ | | | 31.30 | 96.58 | 95.00 | 0.9171 | 0.9459 | 0.9530 | 0.7105 |
| + Render-Inpaint | | ✓ | | _31.48_ | 99.13 | 95.48 | 0.9403 | 0.9663 | 0.9877 | _0.7392_ |
| + Object-consistent | | | ✓ | 31.34 | _99.14_ | **95.92** | **0.9467** | _0.9684_ | _0.9881_ | 0.7385 |
| Full (Ours) | ✓ | ✓ | ✓ | **31.49** | **99.34** | _95.81_ | _0.9463_ | **0.9685** | **0.9885** | **0.7466** |

## 4.3 ABLATION STUDY

We conduct ablation studies using CLIP and VBench score to evaluate the effectiveness of our modules (Table 3, Figures 7 and 8). Our full model demonstrates robust object-level consistency and effectively mitigates truncated objects, incoherent structures, and ghosting artifacts.

**Effect of Geometry-aware Video Depth Refinement.** We evaluate the effectiveness of our refinement module by removing it from the pipeline (denoted as w/o Geometry-aware). As shown in Figure 7, this results in truncated and distorted object structures under camera motion, due to the lack of geometric and temporal consistency. In Table 3, compared with the baseline, CLIP-V improves from 94.80 to 95.00, reflecting stronger view alignment, while Imaging Quality rises from 0.7072 to 0.7105 when the refinement is enabled. Our refinement preserves both geometric and temporal consistency, enabling realistic and stable dynamic scene generation.

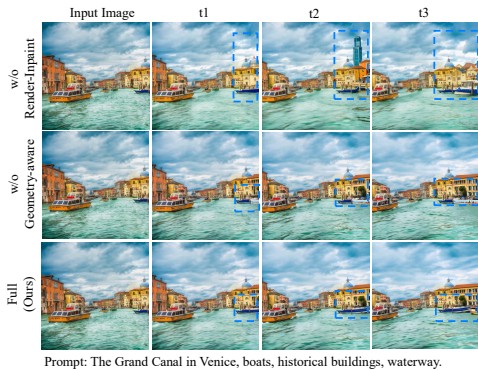

Figure 7: **Ablation study.** Our full model achieves object-level consistency, mitigating truncated objects (second row) and incoherent structures (first row).

**Effect of Rendering-guided Video Inpainting.** Figure 7 shows replacement of our rendering-guided video inpainting module (denoted as *w/o Render-Inpaint* ) with an image inpainting model (Rombach et al., 2022) leads to temporal inconsistencies. In particular, flickering artifacts and structural mismatches of dynamic objects undergoing appearance changes and spatial displacements across frames. This issue is further validated by the VBench scores in Table 3. Compared with the baseline, CLIP-V improves from 94.80 to 95.48, indicating enhanced view alignment, while Imaging Quality increases from 0.7072 to 0.7392 when the rendering-guided video inpainting module is enabled.

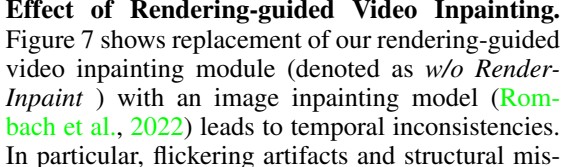

These results highlight the importance of our rendering-guided video inpainting in preserving object consistency and temporal coherence under large viewpoint changes.

**Effect of Object-consistent Temporal Modeling.**
To evaluate the impact of our object-consistent temporal modeling strategy, we replace it with the frame-wise layer composition method from Wonder-World (Yu et al., 2025), which does not incorporate explicit temporal modeling. As illustrated in Figure 8, this replacement introduces temporal artifacts, most notably ghosting where dynamic objects appear duplicated within a single frame. For example, in the first row the boy is rendered twice due to inconsistent object positioning across frames. These artifacts result from the frame-independent nature of WonderWorld's layering, which processes each frame in isolation without temporal context. Consequently, dynamic objects exhibit duplication, ghosting, or abrupt structural shifts. In Table 3, compared with the baseline, object-consistent modeling yields clear gains: Subject Consistency rises from 0.9175 to 0.9467, and Motion Smoothness improves from 0.9530 to 0.9881, when the object-consistent temporal modeling module is enabled. Our object-consistent temporal modeling explicitly tracks both static and dynamic components across frames, maintaining geometrically and temporally coherent dynamic scene generation.



Prompt: *Link taking adventure in a world brimming with magic and wonder, majestic mountains, river, boy.*

Figure 8: **Ablation on object-consistent temporal modeling.** Ghosting and duplication appear without temporal modeling.

## 4.4 USER STUDY

We conducted a user study with approximately 50 anonymous participants to evaluate perceptual quality. Each participant was shown videos generated by our method and three baselines: TrajectoryAttention (Xiao et al., 2025), WonderWorld (Yu et al., 2025), and WonderJourney (Yu et al., 2024a). The videos were presented in random order across seven scene styles. All methods were tested with same input images, text prompts,

Table 4: **User study.** Percentage of participants preferring each method as the best visual result.

| Method | Preference (%) |
|---|---|
| TrajectoryAttention (Xiao et al., 2025) | 12.0 |
| WonderWorld (Yu et al., 2025) | 2.6 |
| WonderJourney (Yu et al., 2024a) | 1.3 |
| WorldCrafter (Ours) | **84.1** |

and camera trajectory for a fair comparison. Participants were asked to select the result they found most visually appealing among the four options. As summarized in Table 4, WorldCrafter achieved a preference of 84.1%, substantially higher than the baselines (12.0%, 2.6%, and 1.3%). These results show that users consistently favor our method for its visual quality, structural fidelity, and temporal coherence. Additional details of user study are provided in Appendix K.

## 5 CONCLUSION

We introduce WorldCrafter, a new framework for interactive dynamic scene generation from a single image that enforces both geometric and temporal consistency. The design integrates geometry-aware depth refinement with object-consistent temporal modeling, reducing artifacts such as texture distortion, ghosting, and temporal flickering while avoiding explicit pixel-wise reprojection. Quantitative and qualitative experiments across diverse scenes demonstrate that WorldCrafter achieves high-fidelity, temporally coherent, and controllable results, establishing a promising direction for single-view interactive dynamic scene synthesis.

**Limitations.** Although our method enables interactive dynamic scene generation and opens new research directions, it cannot model physics-based interactions in complex or high-resolution scenes.

**Broader Impacts.** We will release code and models for reproducibility. This work enables creative and educational applications by supporting controllable dynamic scene generation from a single image. Potential misuse, such as unauthorized media manipulation of interactive media content, should be carefully considered and mitigated in future use.

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

## APPENDIX

### APPENDIX CONTENTS

## A APPENDIX OVERVIEW

We provide a comprehensive appendix to support the proposed WorldCrafter framework. Section B summarizes the compared methods, grouped into *Interactive 3D Scene Generation* and *Camera-controlled Video Generation*. Section C introduces the evaluation metrics, CLIP (Radford et al., 2021) and VBench (Huang et al., 2024), used to evaluate semantic alignment, temporal coherence, and view consistency on the WonderWorld (Yu et al., 2025) dataset. Section D describes the dataset adapted from WonderWorld, while Section E introduces the preliminaries. Section F explains the architecture, and Section G describes the implementation details. Section H provides pseudocode for the core components, *Geometry-aware Video Depth Refinement* and *Object-consistent Temporal Modeling*. Section I presents the details of the ablation studies, Section J presents extended qualitative comparisons, Section K describes a perceptual user study, and Section L discusses failure cases. To support reproducibility, we have uploaded the source code and will release it publicly.

## B COMPARED METHODS

We compare our approach against representative baselines from two categories: *Interactive 3D Scene Generation* and *Camera-Controlled Video Generation*, abbreviated as *3D.* and *Camera.*, respectively, in Table 1 of the main paper.

**Interactive 3D Scene Generation.**

- **WonderJourney** (Yu et al., 2024a): Generates editable 3D scenes from single images or text prompts via point cloud reconstruction. It supports user-guided editing and scene composition but lacks real-time interaction and efficient rendering.
- **WonderWorld** (Yu et al., 2025): Extends WonderJourney by enabling interactive 3D scene editing and accelerating rendering with the Fast LAyered Gaussian Surfels (FLAGS) representation. However, it remains limited in generalization to dynamic scenes and diverse camera motions.

**Camera-Controlled Video Generation.**

- **TrajectoryCrafter** (YU et al., 2025): Proposes a dual-stream diffusion framework for monocular video redirection conditioned on camera trajectories. It performs well in controlled settings but struggles with large viewpoint shifts.
- **TrajectoryAttention** (Xiao et al., 2025): Introduces trajectory-aware attention mechanisms for fine-grained, motion-conditioned video generation. While it improves camera control, it shows instability under noisy or ambiguous trajectories.
- **ReCamMaster** (Bai et al., 2025): Adapts text-to-video generation models for novel-view video synthesis, showing strong results in synthetic environments but failing to generalize to real-world data due to limited motion diversity and insufficient occlusion handling.

These baselines cover both interactive 3D scene and camera-conditioned video generation, providing a comprehensive comparison for evaluating our proposed framework.

## C EVALUATION METRICS

We evaluate scene generation methods using **CLIP** (Radford et al., 2021) and **VBench** (Huang et al., 2024), measuring semantic alignment, temporal coherence, and view consistency. All methods are tested on the WonderWorld (Yu et al., 2025) dataset for fair comparison.

**CLIP-based Metrics.**

- **CLIP-T (Text-to-Frame Alignment):** Measures the average CLIP similarity between each video frame and its text prompt, reflecting how well the visual content aligns with the description.
- **CLIP-F (Temporal Coherence):** Computes the average CLIP similarity between consecutive video frames, capturing temporal smoothness and visual continuity.
- **CLIP-V (View Consistency):** Evaluates the CLIP similarity between source and generated frames at the same timestamp, indicating consistency across camera trajectories.

**VBench Evaluation Protocols.**

- **Subject Consistency**: Evaluates the temporal consistency of the main subject's appearance, ensuring attributes such as identity, color, shape, and texture are preserved across frames. Failures may lead to identity drift or artifacts that reduce realism.

- **Background Consistency**: Evaluates the coherence of background elements, checking whether lighting, structure, and texture remain stable across frames. Inconsistencies reduce visual quality and expose weaknesses in the model's ability to generate stable videos.

- **Motion Smoothness**: Evaluates the continuity of motion, verifying that trajectories are realistic and free of jitter. Smooth motion enhances realism and reflects the model's ability to generate temporally coherent sequences.

- **Aesthetic Quality**: Evaluates the visual appeal of video frames, considering composition, color harmony, and overall impression. High aesthetic quality reflects the model's ability to generate content aligned with human preferences.

- **Imaging Quality**: Measures the technical fidelity of video frames, focusing on resolution, sharpness, noise, and the absence of artifacts. High imaging quality reflects the model's capacity to generate clear and stable visuals.

These metrics jointly offer a comprehensive assessment of scene consistency, visual fidelity, and temporal stability in dynamic scene generation. Higher scores reflect stronger performance.

## D   DATASET

We evaluate WorldCrafter and all baselines on the WonderWorld dataset (Yu et al., 2025), which includes seven diverse scenes spanning nature, city, fantasy, and campus categories. To simulate camera motion, we apply predefined trajectories such as panning (left, right, up, down), zooming (in, out), and rotations of $\pm 20°$, $\pm 40°$, and $\pm 60°$. These motion patterns generate dynamic scenes for a comprehensive evaluation of model performance.

## E   PRELIMINARIES

**Gaussian Splatting.** 3D Gaussian Splatting (3DGS) (Kerbl et al., 2023) represents scenes using explicit 3D representation for efficient reconstruction and novel view synthesis. Unlike implicit methods such as NeRF (Mildenhall et al., 2020), 3DGS (Kerbl et al., 2023) directly projects Gaussians into image space via a differentiable splatting pipeline to support fast optimization and real-time rendering. Each 3D Gaussian point is defined by several key attributes: spatial position $\mathcal{X} \in \mathbb{R}^3$, color represented as spherical harmonics (SH) $\mathcal{C} \in \mathbb{R}^k$, opacity $\alpha \in \mathbb{R}$, orientation encoded as a quaternion $r \in \mathbb{R}^4$, and scale $s \in \mathbb{R}^3$. $k$ denotes the number of SH functions. To render an image, each pixel aggregates contributions from multiple Gaussians. When multiple Gaussians influence the same pixel, the final color is obtained by compositing the $N$ ordered contributions as follows:

$$C = \sum_{i \in N} c_i \alpha_i \prod_{j=1}^{i-1}(1 - \alpha_j), \tag{6}$$

where $c_i$ and $\alpha_i$ denote the color and opacity of the $i$-th Gaussian, respectively. In our framework, 3DGS provides explicit spatial rendering capabilities for interactive dynamic scene generation with high-quality dynamic composition and fast photorealistic rendering.

## F   ARCHITECTURE

WorldCrafter adopts an image-to-scene pipeline that synthesizes an initial video from a single input image and text prompt (See Figure 3). The process begins with an Image-to-Video model (e.g., CogVideoX), which generates initial frames used as reference inputs for subsequent refinement. To enhance geometric fidelity, a depth estimator produces per-frame depth maps, which are refined by the geometry-aware depth module. This refinement enforces intra-segment consistency guided by

---

**Algorithm 1** Interactive Dynamic Scene Generation (WorldCrafter)

---

**Input:** Initial image $\mathbf{I}_0$, initial prompt $\mathcal{P}_0$
**Output:** Dynamic scenes $\mathcal{W} = \{\mathcal{W}_0, \mathcal{W}_1, \ldots, \mathcal{W}_{T-1}\}$
**Runtime Output:** Accumulated rendered video $\mathbf{V}_{\text{rend}}$
**Runtime User Interaction:** Camera pose $\mathbf{C}_t$, text prompt $\mathcal{P}_t$

1: $\mathbf{C}_0 \leftarrow \text{Identity}(4 \times 4)$       ▷ Initialize camera pose
2: $\mathbf{M}_0 \leftarrow \mathbf{1}^{H \times W}$       ▷ Initial scene mask (fully visible)

3: $\mathbf{V}_0 \leftarrow \text{ImageToVideo}(\mathbf{I}_0, \mathcal{P}_0)$       ▷ Generate initial video
4: $\mathbf{D}_0 \leftarrow \text{VideoDepthEstimator}(\mathbf{V}_0)$       ▷ Estimate depth
5: $\mathbf{D}_0^{\text{refined}} \leftarrow \text{GeometryAwareRefiner}(\mathbf{D}_0)$       ▷ Refine geometry
6: $\mathcal{W}_0 \leftarrow \text{ObjectConsistentModeling}(\mathbf{V}_0, \mathbf{D}_0^{\text{refined}}, \mathcal{P}_0, \mathbf{C}_0)$       ▷ Initialize dynamic scene
7: $\mathcal{W} \leftarrow \{\mathcal{W}_0\}$
8: $\mathbf{V}_{\text{rend}} \leftarrow \text{Render}(\mathcal{W}_0, \mathbf{C}_0)$       ▷ Initial rendering

9: **for** $t = 0$ to $T - 1$ **do**
10:     Observe current scene $\mathcal{W}_t$
11:     Receive user input: text prompt $\mathcal{P}_{t+1}$ and camera pose $\mathbf{C}_{t+1}$
12:     $\mathbf{M}_{t+1} \leftarrow \mathbf{M}_t$       ▷ Keep mask fixed across time
13:     $\mathbf{Z}_{t+1} \leftarrow \mathbf{W}_{\text{update}}(\mathcal{W}_t, \mathbf{M}_{t+1}, \mathcal{P}_{t+1}, \mathbf{C}_{t+1})$       ▷ Update latent state
14:     $\mathcal{W}_{t+1} \leftarrow \mathbf{G}_{\text{world}}(\mathbf{Z}_{t+1})$       ▷ Generate new dynamic scene
15:     $\mathcal{W} \leftarrow \mathcal{W} \cup \{\mathcal{W}_{t+1}\}$       ▷ Append to sequence
16:     $\mathbf{V}_{\text{rend}} \leftarrow \mathbf{V}_{\text{rend}} \,\|\, \text{Render}(\mathcal{W}_{t+1}, \mathbf{C}_{t+1})$       ▷ Accumulate rendered video
17: **end for**
18: **return** $\mathcal{W}$

---

semantic masks, suppressing artifacts such as distorted boundaries or flying points. The refined depths are then processed by the object-consistent temporal modeling module, which back-projects pixels into 3D, groups them into segment-conditioned Gaussian primitives, and renders novel views under the current camera pose. This stage maintains spatial-temporal coherence by explicitly tracking objects across frames and detecting occlusions via a geometry-aware visibility mask. Finally, the rendered frames and masks are combined with text embeddings from a pretrained encoder, which guide a rendering-aware inpainting network to fill missing regions and align content with user-provided descriptions. The outputs are decoded into novel-view dynamic scenes with stable geometry, coherent motion, and preserved semantics. Our framework follows the interactive setting of previous works (e.g., WonderWorld (Yu et al., 2025) and WonderJourney (Yu et al., 2024a)) but extends it with explicit geometry-aware refinement and object-consistent modeling, enabling controllable and realistic dynamic scene generation.

## G IMPLEMENTATION DETAILS

All experiments are conducted on an NVIDIA RTX 6000 GPU with 46 GB memory. Following WonderWorld (Yu et al., 2025), we use single input images and associated text prompts to simulate six camera motions (pan left/right/up/down, zoom in/out) via the official inference pipelines of each baseline for fair comparison. Our method extends the static 3D framework of WonderWorld by incorporating CogVideoX-Fun-5B (Yang et al., 2024) for video diffusion (output resolution: 512×512) and DepthCrafter (Hu et al., 2024) for generating temporally consistent depth maps. We evaluate performance both quantitatively and qualitatively across all motion types.

## H ALGORITHMS

### H.1 WORLDCRAFTER ALGORITHM OVERVIEW

We present the pseudo-code of the proposed *WorldCrafter* framework in Algorithm 1, which enables dynamic scene generation from a single initial image and text prompt. The framework consists of two stages: initialization and iterative scene generation.

In the initialization stage, given a single input image $\mathbf{I}_0$ and text prompt $\mathcal{P}_0$, WorldCrafter synthesizes an initial video $\mathbf{V}_0$ using an image-to-video model (Wang et al., 2025a; Team., 2025). A corresponding depth map $\mathbf{D}_0$ is estimated with a video depth estimator (Hu et al., 2025) and refined to ensure spatial consistency. An object-consistent temporal modeling module then integrates the

video, refined depth, and initial camera pose $\mathbf{C}_0$ to construct the initial dynamic scene $\mathcal{W}0$, which is rendered to produce the output video $\mathbf{V}_{\text{rend}}$.

In the iterative stage, the user provides updated text prompts $\mathcal{P}_{t+1}$ and camera poses $\mathbf{C}_{t+1}$ at each time step. A latent updater integrates the current world $\mathcal{W}_t$, a fixed spatial mask $\mathbf{M}_t$, and the user inputs to compute a latent representation $\mathbf{Z}_{t+1}$, which is decoded into the next dynamic scene $\mathcal{W}_{t+1}$. The new scene is rendered and appended to the accumulated output $\mathbf{V}_{\text{rend}}$.

## H.2 GEOMETRY-AWARE VIDEO DEPTH REFINEMENT

As shown in Algorithm 2, the refinement employs binary semantic masks from SAM (Kirillov et al., 2023) to partition each frame into coherent regions. A filtering operator (e.g., median filtering) is then applied within each region to replace noisy depth values with a region-level statistic, while pixels outside the masks retain their original depth. This process produces spatially consistent depth maps that support stable rendering and dynamic scene generation.

---

**Algorithm 2** Geometry-aware Video Depth Refinement

---

**Input:** Video frames $\mathbf{V} = \{\mathbf{I}_t\}_{t=1}^T$
**Output:** Refined depth maps $\hat{\mathbf{D}} = \{\hat{D}_t\}_{t=1}^T$

1: **for** $t = 1$ to $T$ **do**
2:    $D_t \leftarrow \text{DepthEstimator}(\mathbf{I}_t)$     ▷ Initial depth from video depth estimator (Hu et al., 2025)
3:    $\mathcal{S} = \{\mathcal{S}_j\}_{j=1}^{N_s} \leftarrow \text{SAM}(\mathbf{I}_t)$     ▷ Semantic masks from SAM (Kirillov et al., 2023)
4:    Initialize $\hat{D}_t \leftarrow \mathbf{0}^{H \times W}$     ▷ Refined depth map
5:    **for** $j = 1$ to $N_s$ **do**
6:       $M_j \leftarrow \mathcal{S}_j$     ▷ Binary mask for segment $j$
7:       $v_j \leftarrow \text{median}\{D_t(x,y) \mid M_j(x,y) = 1\}$     ▷ Segment-wise depth median
8:       **for all** pixels $(x,y)$ where $M_j(x,y) = 1$ **do**
9:          $\hat{D}_t(x,y) \leftarrow v_j$     ▷ Assign median to all pixels in segment
10:       **end for**
11:    **end for**
12:    **for all** pixels $(x,y)$ not covered by any $M_j$ **do**
13:       $\hat{D}_t(x,y) \leftarrow D_t(x,y)$     ▷ Keep original depth
14:    **end for**
15: **end for**
16: **return** $\hat{\mathbf{D}}$

---

## H.3 OBJECT-CONSISTENT TEMPORAL MODELING

As illustrated in Algorithm 3, the object-consistent temporal modeling module integrates geometric cues, semantic segmentation, and temporal memory to maintain coherence across frames. For each frame, semantic masks from SAM (Kirillov et al., 2023) guide the grouping of pixels into object-level regions, which are back-projected into 3D and represented as Gaussian primitives. These representations are accumulated over time and rendered from the current camera pose to obtain novel-view candidates. An occlusion-aware visibility mask $M_t$ is then derived from the rendered geometry to identify unreliable regions. Finally, a prompt-guided inpainting network $\mathcal{I}_{\text{net}}(\tilde{I}_t^{\text{view}}, M_t, \mathcal{P})$ fills in the masked areas, and the temporal memory $\mathcal{M}$ is updated to preserve object consistency across the sequence. This design yields temporally coherent frames while ensuring object-level alignment for dynamic scene generation.

# I ABLATION DETAILS

We provide additional explanations for the ablation experiments reported in Table 3 of the main paper. All variants are evaluated under the VBench (Huang et al., 2024) protocol to evaluate the contributions of each component in our framework. Ablations are conducted by enabling specific modules, with each row in Table 3 showing the modules used (✓). CLIP scores are reported on a scale from 0 to 100, with other metrics normalized to [0,1]. Settings are summarized below:

---

**Algorithm 3** Object-consistent Temporal Modeling

---

**Input:** Video Frames $\mathbf{V} = \{\mathbf{I}_t\}_{t=1}^T$,
Depth Maps $\mathbf{D} = \{D_t\}_{t=1}^T$,
Camera Pose $\mathbf{C} = \{C_t = (K_t, R_t, T_t)\}_{t=1}^T$,
Text Prompt $\mathcal{P}$

**Output:** Final Dynamic Scenes $\hat{\mathbf{V}} = \{\hat{I}_t\}_{t=1}^T$

1: Initialize memory buffer $\mathcal{M} \leftarrow \emptyset$             ▷ Temporal memory for object tracking
2: **for** $t = 1$ to $T$ **do**
3:     $\mathcal{S} = \{\mathcal{S}_j^t\}_{j=1}^{N_s} \leftarrow \text{SAM}(\mathbf{I}_t)$                    ▷ Semantic masks
4:     Initialize $\mathcal{G}_t \leftarrow \emptyset$            ▷ Set of 3D Gaussians for current frame
5:     **for** $j = 1$ to $N_s$ **do**
6:        **for all** pixels $(x, y)$ where $\mathcal{S}_j^t(x, y) = 1$ **do**
7:           $\mathbf{P}_{t,j}^{3D}(x, y) \leftarrow R_t^{-1}\left(D_t(x, y) \cdot K_t^{-1}[x, y, 1]^T - T_t\right)$     ▷ Backproject pixel to 3D
8:           $(\mu_i^j, \Sigma_i^j, \mathbf{c}_i^j, \alpha_i^j) \leftarrow \text{EncodeGaussian}(\mathbf{P}_{t,j}^{3D}, \mathbf{I}_t(x, y))$
9:           Add Gaussian to $\mathcal{G}_t^j$
10:        **end for**
11:        Add $\mathcal{G}_t^j$ to $\mathcal{G}_t$
12:     **end for**
13:     $\tilde{I}_t^{\text{view}} \leftarrow \mathcal{R}_{\text{3DGS}}\left(\bigcup_{s=1}^{t-1} \mathcal{G}_s \cup \mathcal{G}_t, \ C_t\right)$     ▷ Render from temporally accumulated Gaussians
14:     $M_t \leftarrow \text{VisibilityMaskFromGeometry}(\tilde{I}_t^{\text{view}}, D_t)$     ▷ Binary occlusion-aware mask
15:     $\hat{I}_t \leftarrow \mathcal{I}_{\text{net}}(\tilde{I}_t^{\text{view}}, M_t, p)$           ▷ Rendering-guided inpainting using prompt
16:     Add $\hat{I}_t$ to $\hat{\mathbf{V}}$
17:     $\mathcal{M} \leftarrow \text{UpdateTemporalMemory}(\mathcal{M}, \mathcal{S}, \mathcal{G}_t)$     ▷ Track segments and update memory
18: **end for**
19: **return** $\hat{\mathbf{V}}$

---

- **Baseline**: Serves as the minimal configuration without geometry-aware depth refinement, rendering-guided inpainting, or object-consistent temporal modeling. This setting exhibits severe structural distortions, temporal flickering, and inconsistent object appearances.

- **+ Geometry-Aware Depth Refinement**: Incorporating geometry-aware refinement alleviates truncated or deformed structures during camera motion, improving overall scene stability. However, temporal inconsistencies remain due to the absence of inpainting and temporal modeling.

- **+ Rendering-Guided Inpainting**: Adding the rendering-guided video inpainting module (instead of an image-only alternative (Rombach et al., 2022)) provides temporally coherent texture completion. This reduces flickering artifacts and improves visual continuity of dynamic objects.

- **+ Object-Consistent Temporal Modeling**: Introducing object-consistent temporal modeling further enforces identity and state consistency of moving entities. Compared with frame-independent composition strategies (e.g., WonderWorld (Yu et al., 2025)), this component suppresses ghosting artifacts such as object duplication.

- **Full (Ours)**: Combines all three components (✓ in all columns), delivering geometry-preserving, temporally stable, and object-consistent video generation across diverse camera trajectories.

## J  QUALITATIVE COMPARISON AND ANALYSIS

We present additional qualitative comparisons in Figures 11–13, focusing on camera-controlled video generation. Our method produces geometry-aware scenes that remain stable across frames, while prior approaches (Xiao et al., 2025; YU et al., 2025) relying on pixel-wise reprojection often suffer from distortions or ghosting under large viewpoint changes. Figures 14 and 15 compare interactive 3D scene models (Yu et al., 2024a; 2025). WorldCrafter better keeps temporal coherence, yielding smoother and more consistent interactions in dynamic environments. Overall, WorldCrafter generates coherent, high-fidelity dynamic scenes that preserve input semantics and appearance, even under complex motions and camera changes.

## K    USER STUDY

We conducted a user study on the questionnaire platform to evaluate perceptual preferences in video quality and temporal consistency, with approximately 50 participants anonymously recruited. Each participant viewed a set of videos generated by our method and three baselines: TrajectoryAttention (Xiao et al., 2025), WonderWorld (Yu et al., 2025), and WonderJourney (Yu et al., 2024a). The videos were presented in randomized order to avoid bias. For fair comparison, each set was generated in seven distinct styles from the same input image and text prompt. As shown in Figure 10, participants were instructed:"Please compare the four videos below and select the one with the best overall quality." They selected a single preferred video for each comparison, considering visual fidelity, geometric accuracy, and temporal coherence. This study design directly captures user preferences across diverse styles and motion scenarios, providing perceptual validation of our method's effectiveness.

## L    FAILURE CASE ANALYSIS

Figure 9 shows representative failure cases that illustrate the limitations of our approach. In particular, the method struggles with fine-grained details such as dynamic water surfaces (e.g., spring water), leading to artifacts like unrealistic shadows. These observations suggest potential future directions, including incorporating physically grounded models of real-world environments.

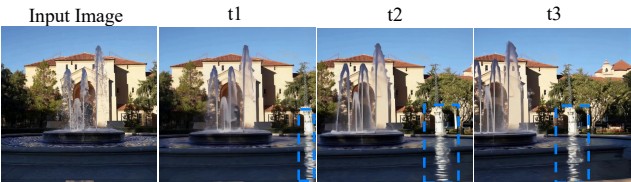

Prompt: *Elegant building, dynamic fountain set, Mediterranean building, campus setting.*

Figure 9: **Failure case under dynamic water and sunlight.** Our method generates physically inconsistent water shadows, reflecting the difficulty of modeling complex interactions between moving water surfaces and sunlight.

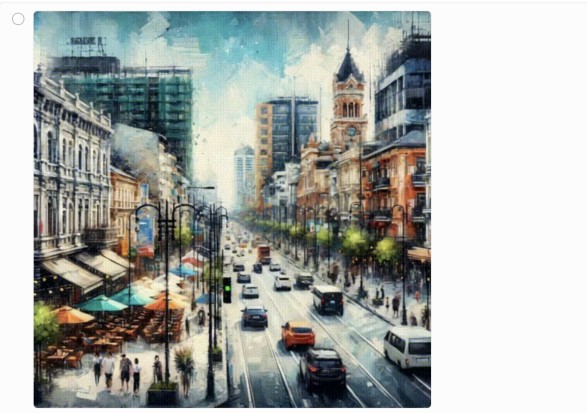

Figure 10: Screenshot of the user study interface.

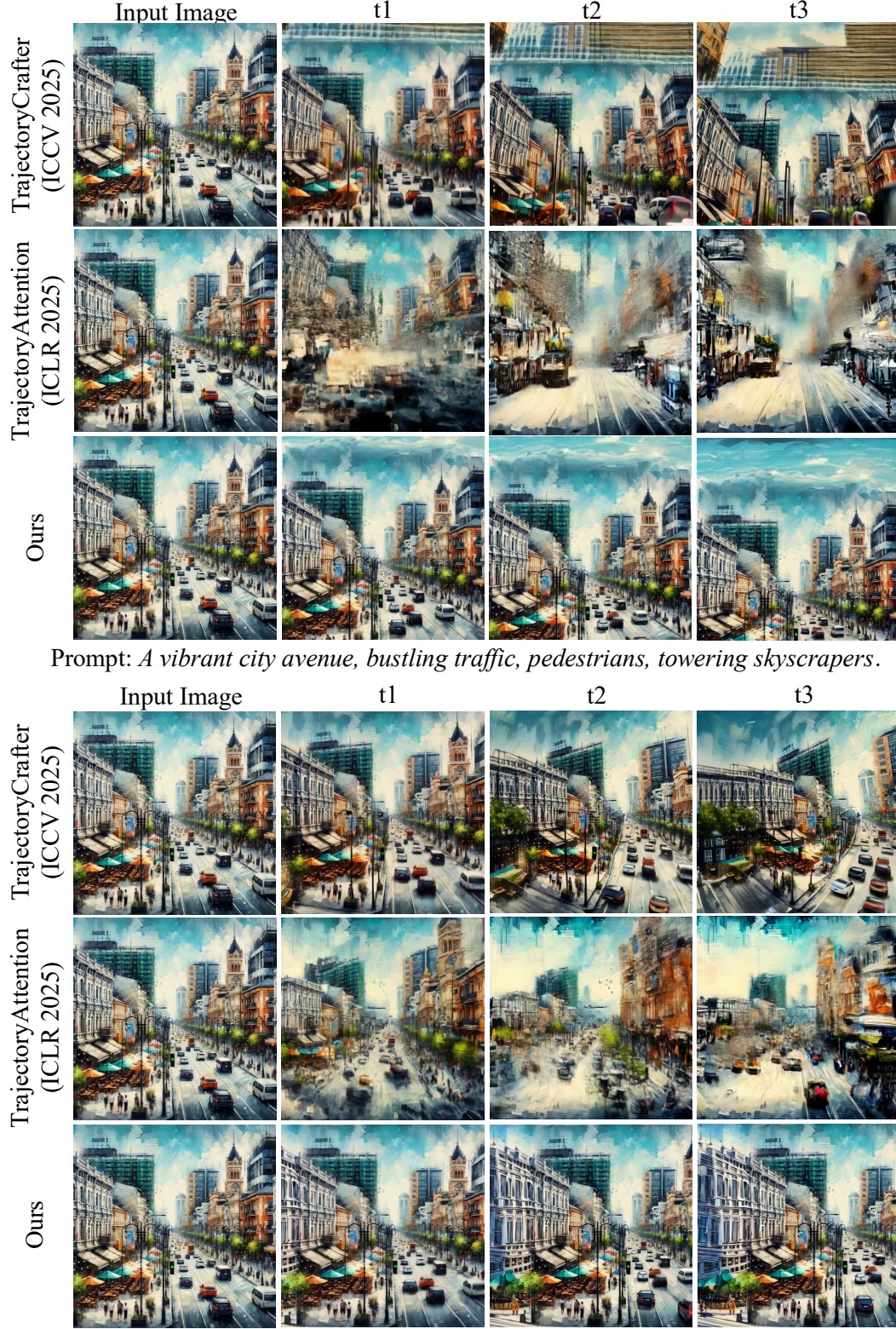

Figure 11: **Comparison with camera-controlled video generation.** Previous methods lack geometry and temporal constraints, causing distortions. WorldCrafter ensures consistency with the input image and prompt. (Top: Pan-Up, Bottom: Pan-Left)

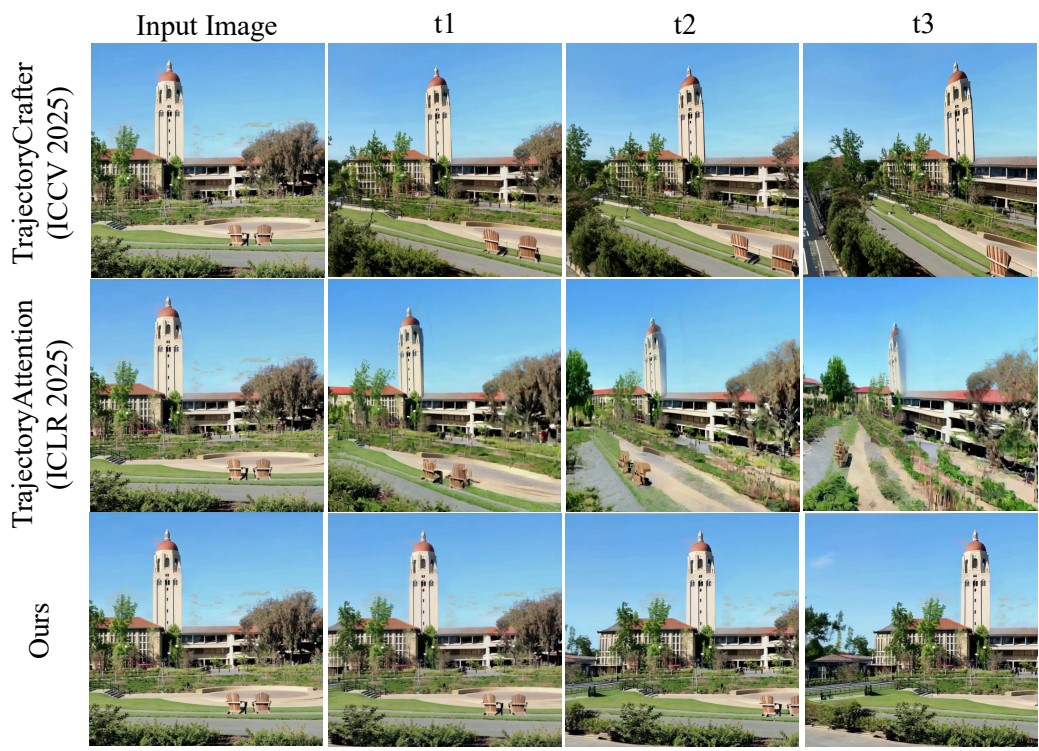

Prompt: *Historic university building, prominent bell tower, lush green lawn, blue sky.*

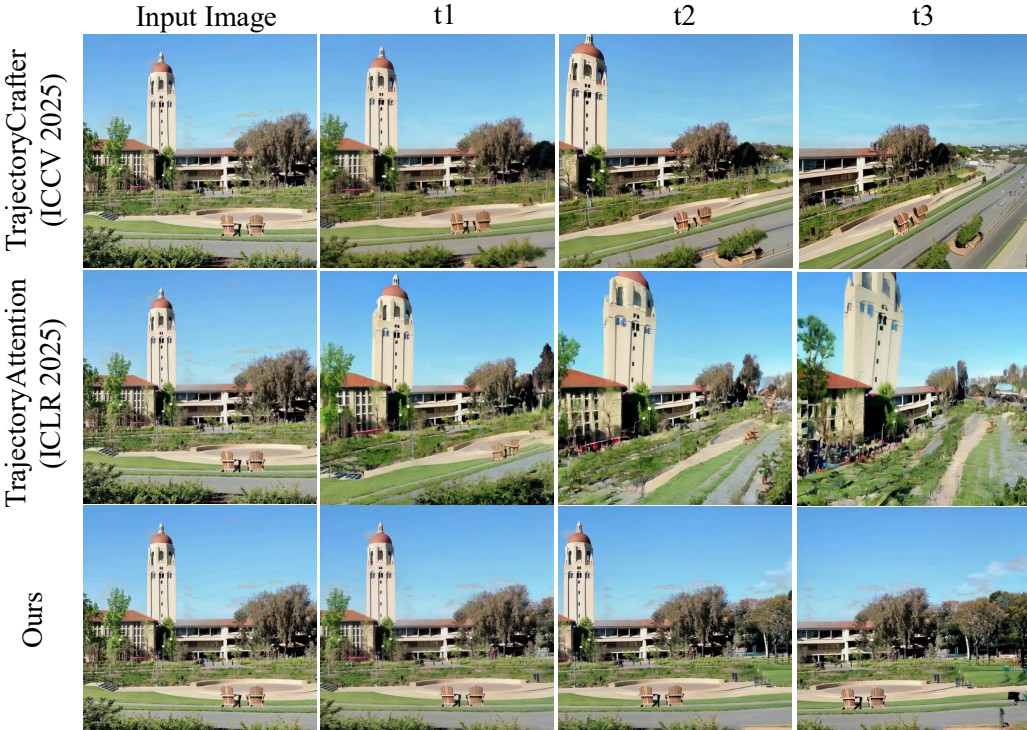

Prompt: *Historic university building, prominent bell tower, lush green lawn, blue sky.*

Figure 12: **Comparison with camera-controlled video generation.** Existing methods suffer from distortions due to missing geometric and temporal constraints. WorldCrafter maintains consistency with the input image and prompt. (Top: Pan-Left, Bottom: Pan-Right)

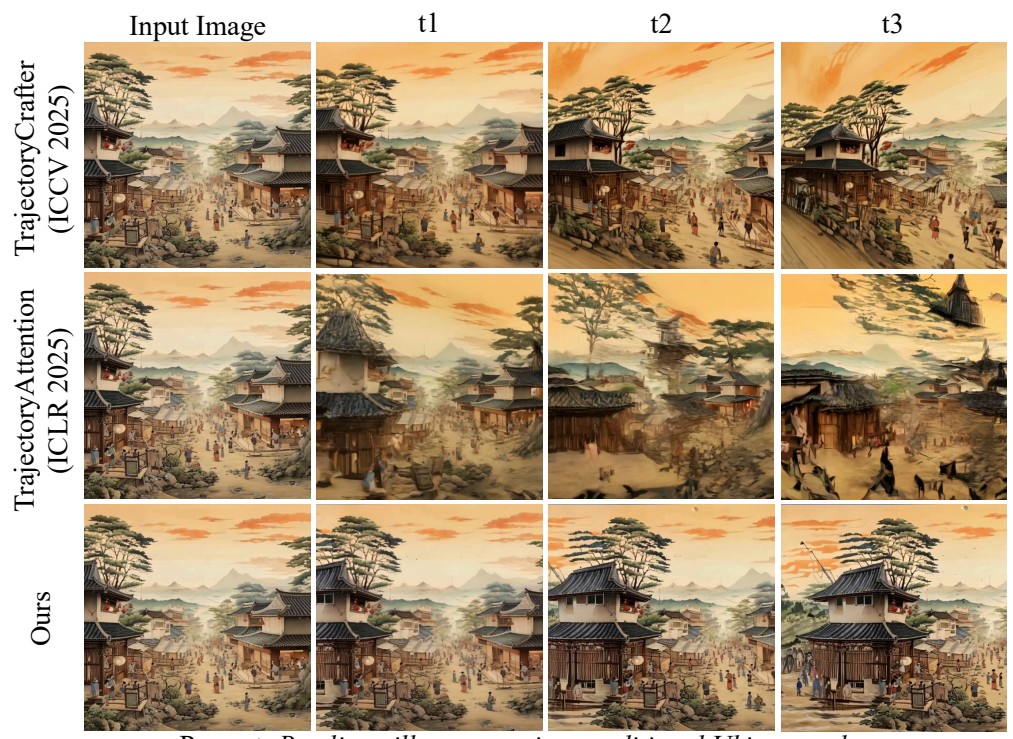

Prompt: *Bustling village scene in a traditional Ukiyo-e style*
*depicting daily life with a backdrop of mountains and sea.*

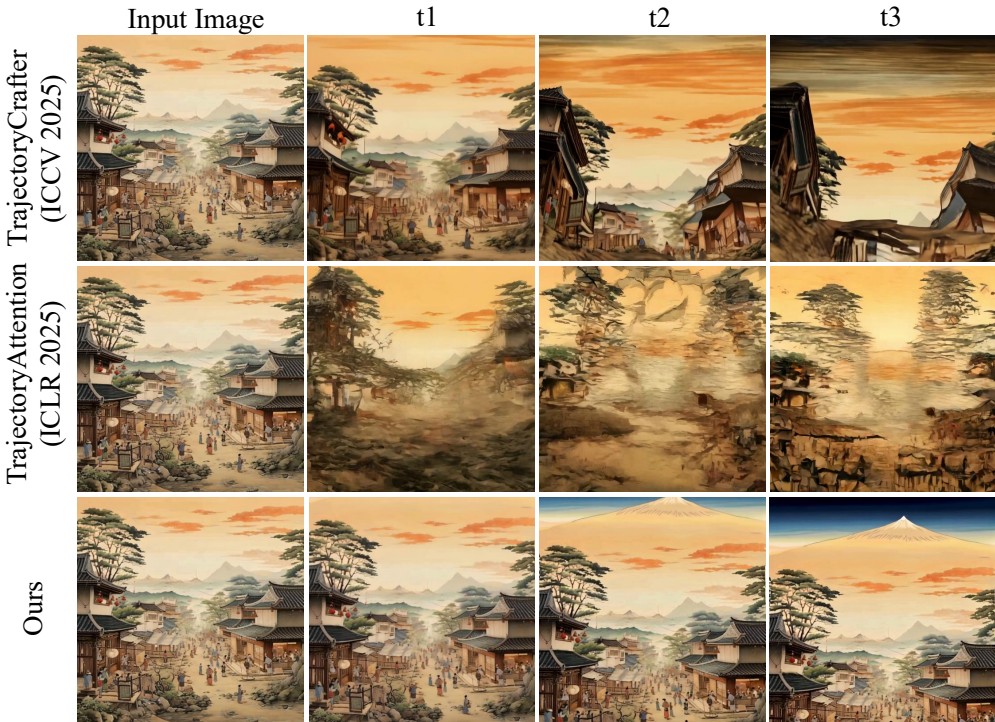

Prompt: *Bustling village scene in a traditional Ukiyo-e style*
*depicting daily life with a backdrop of mountains and sea.*

Figure 13: **Comparison with camera-controlled video generation.** Existing methods yield distortions due to limited geometric and temporal constraints. WorldCrafter maintains consistency with the input image and prompt. (Top: Pan-Left, Bottom: Pan-Up)

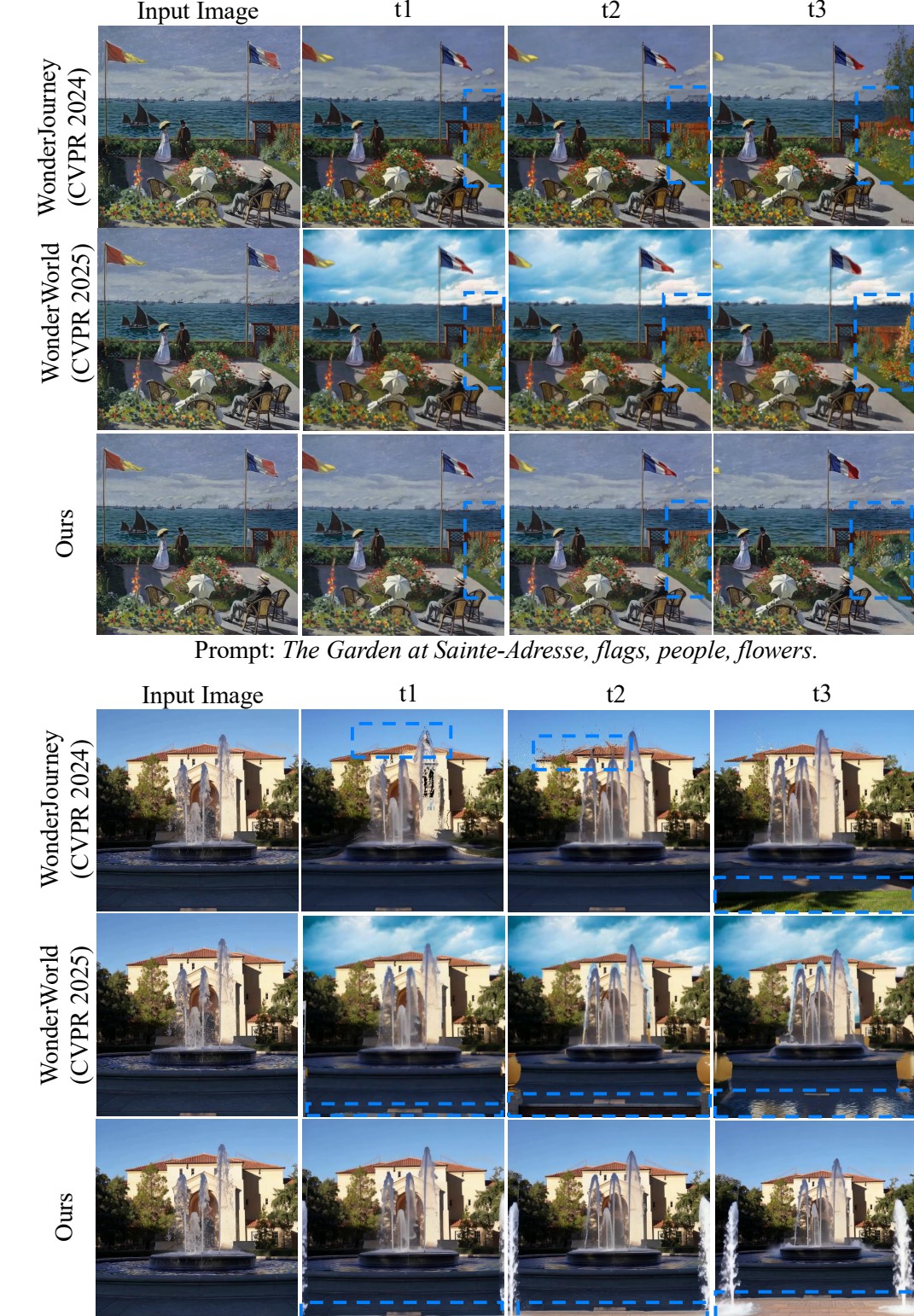

Figure 14: **Comparison with 3D interactive models over time.** Traditional methods show object drift due to missing coherence modeling. WorldCrafter ensures spatial-temporal consistency within dynamic scene. (Top: Pan-Right, Bottom: Pan-Down)

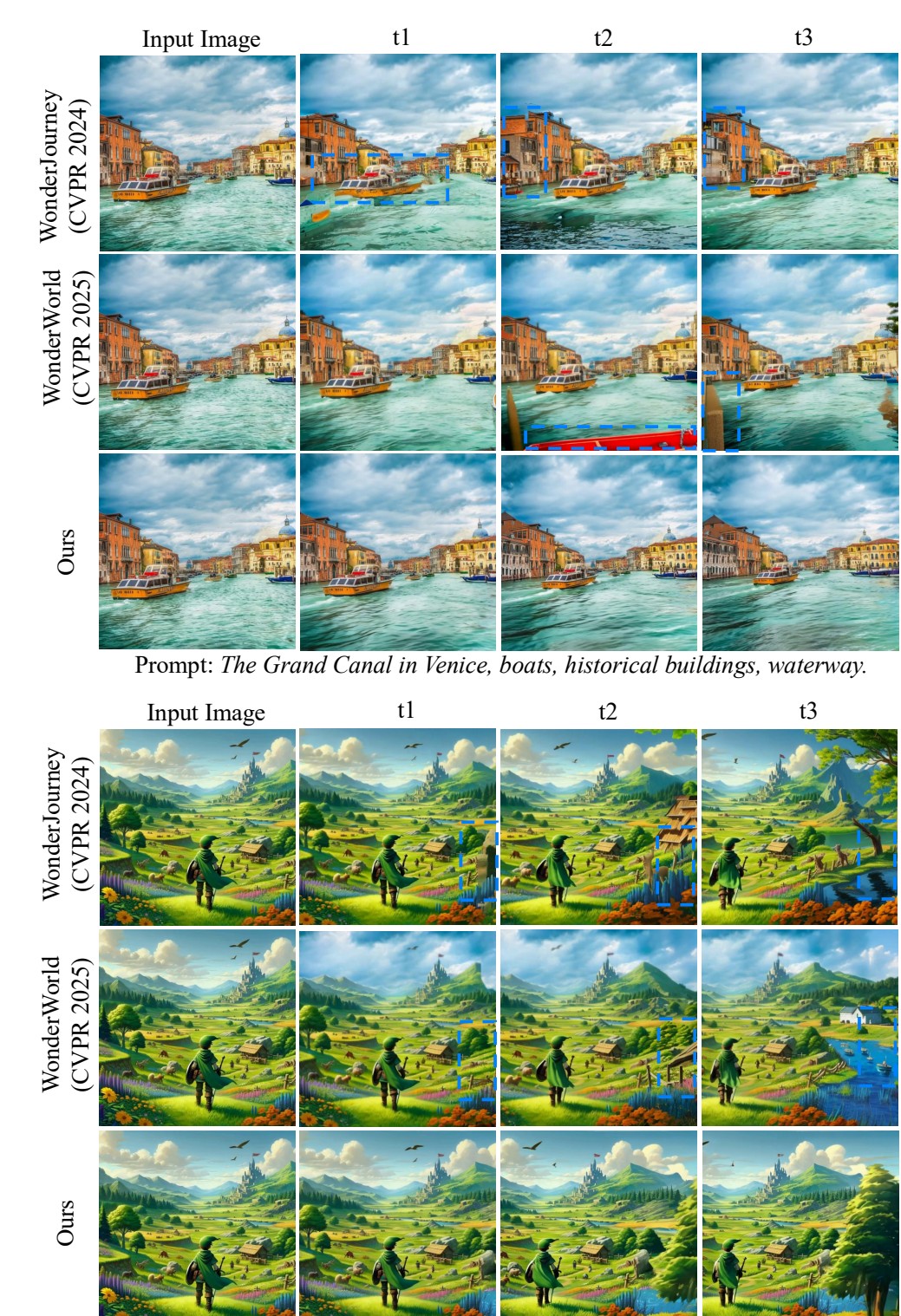

Figure 15: **Comparison with 3D interactive models over time.** Previous methods show temporal inconsistencies due to lack of object-level coherence. WorldCrafter keep spatial-temporal consistency in dynamic scene. (Top: Pan-Left, Bottom: Pan-Right)

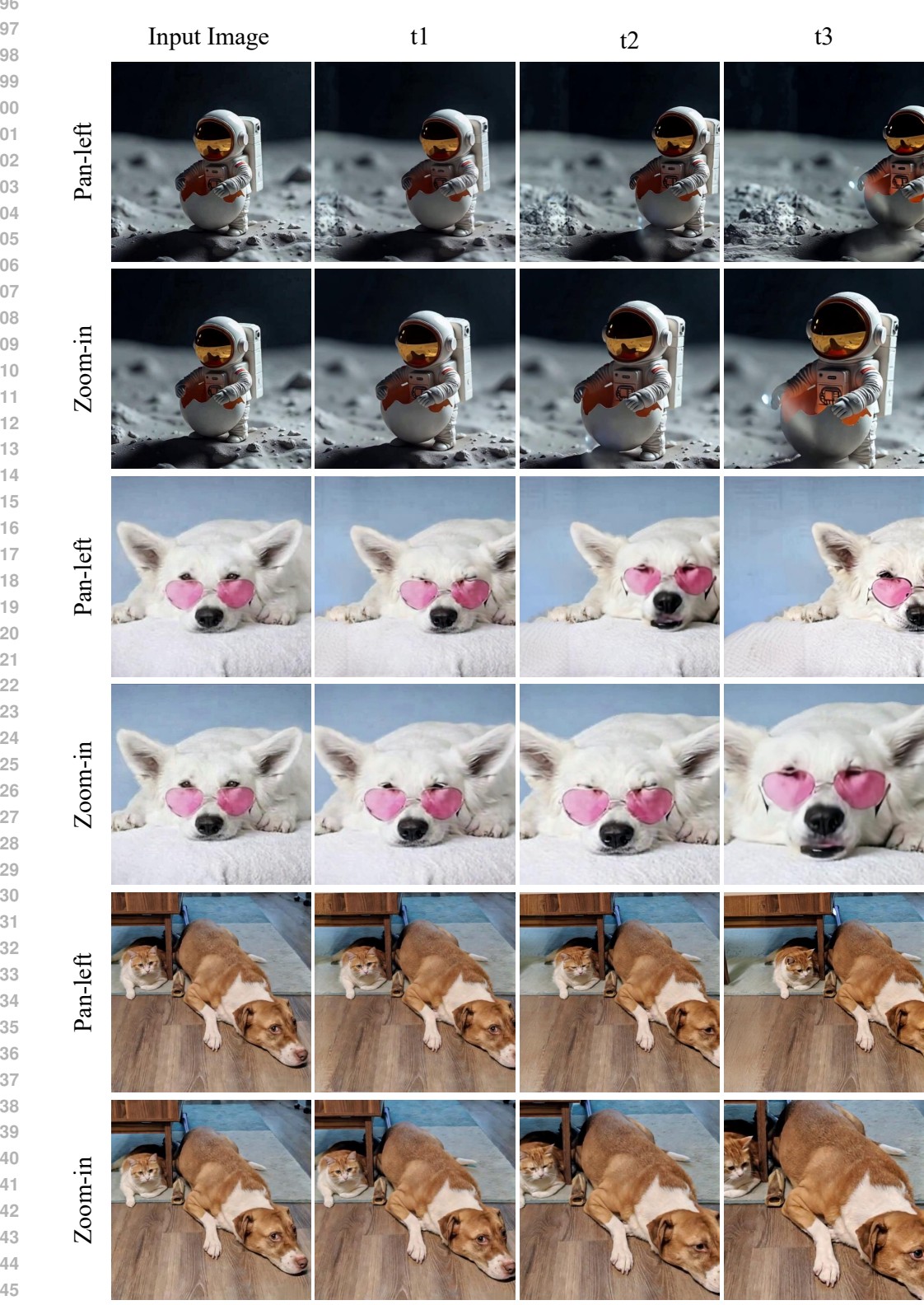

Figure 16: Qualitative results of the proposed WorldCrafter on in-the-wild images. Our framework generates temporally coherent dynamic scenes under diverse camera trajectories (pan and zoom) while preserving object consistency and realism.

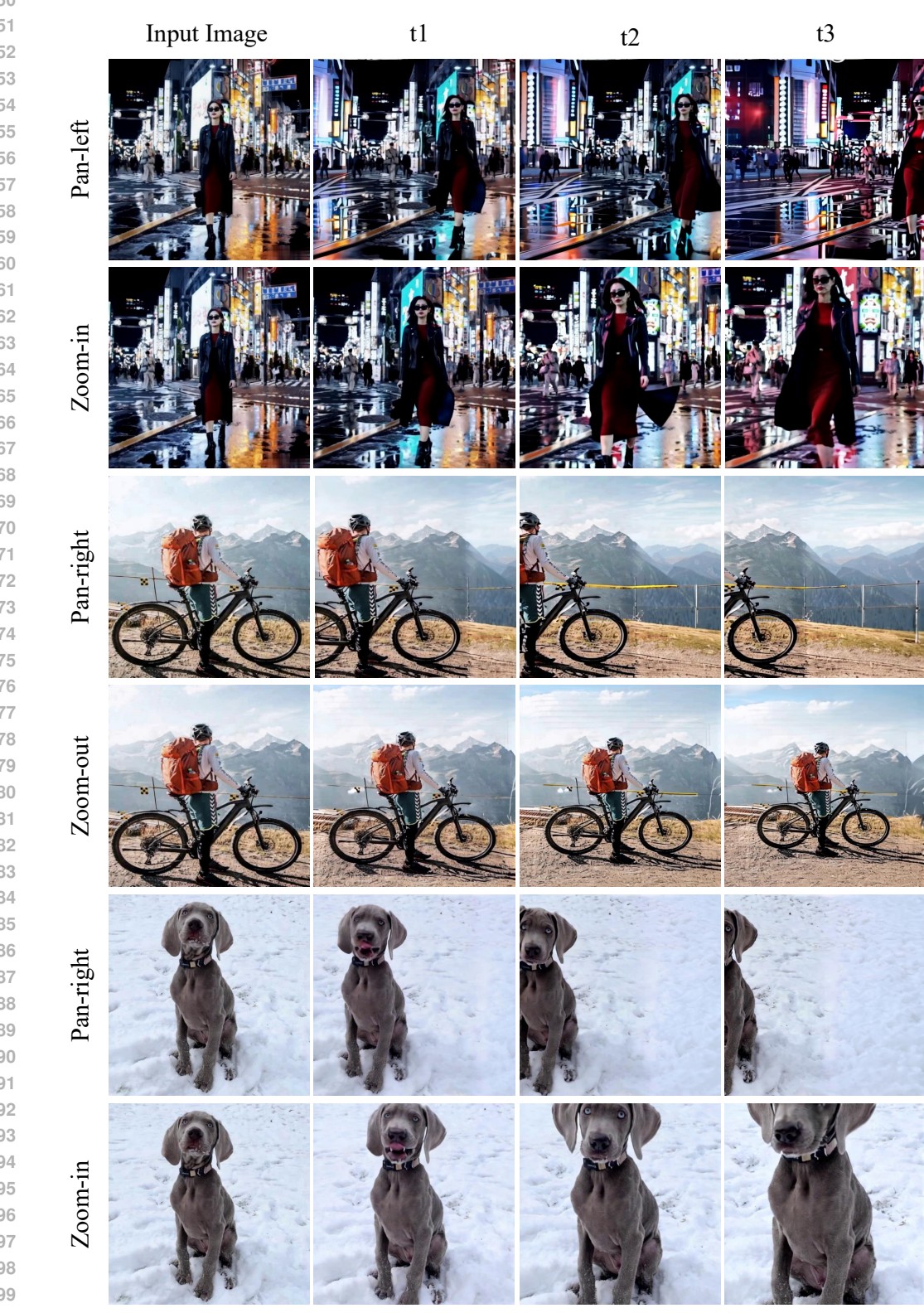

Figure 17: Qualitative results of the proposed WorldCrafter on in-the-wild images. Our framework generates temporally coherent dynamic scenes under diverse camera trajectories (pan and zoom) while preserving object consistency and realism.

