# OpenReview forum: "WorldCrafter: Dynamic Scene Generation from a Single Image with Geometric and Temporal Consistency"
_ICLR.cc/2026/Conference — ICLR 2026 Conference Withdrawn Submission_

### Official Review · Reviewer_dk65 · 2025-10-26

**Soundness:** 3
**Presentation:** 2
**Contribution:** 2
**Rating:** 2
**Confidence:** 4

**Summary:**

In this paper, the authors present a framework that generate novel view scene from a single image by their proposed geometry-aware and temporal modeling. Specifically, they introduce two modules, a geometry-aware video depth refinement and a object-consistent temporal modeling mechanism.

**Strengths:**

- The structure of this paper is complete.
- The authors provide source code, which should be encouraged.

**Weaknesses:**

- Unfair settings for 3D methods. From the supplementary video, it can be seen that the 3D methods exhibit significant flickering in the newly filled areas. The correct approach should be to maintain the filling result of the i-th frame when filling in the i+1 frame, rather than completely refilling every frame from scratch.
- Comparisons with 4D method. The authors mainly compare 3D methods, but currently there are many 4D methods that can be compared, such as Free4D[1], CAT4D [2], GenXD [3], and DimensionX [4].
-  Wrong claims. The authors mentioned that semantic regions share similar depth distributions, but for a river, there is a significant difference in the depth of its foreground and background. Therefore, using Median Filtering may is not a good approach. In addition, Median Filtering sacrifices a lot of fine variations in real geometry (such as slopes/roads with depth gradients, thin structures, and surfaces), making it easy to flatten non planar surfaces into segmented parallel ones.
-  The effectiveness of semantic segmentation. The authors use SAM in both Geometry-aware Video Depth Refinement and Object-consistent Temporal Modeling. But this will introduce more complexity, constraints, and limitations. The author should present and analyze more intermediate results to demonstrate the effectiveness of incorporating semantic segmentation. For example, for a scene with hundreds or thousands of people, how should semantic segmentation be performed
-  Writing. This paper contains a lot of repetitive narratives in the same paragraph. For example, in line 79 to line 95, the authors repeatedly talk about the geometry-aware video depth refinement and the object-consistent temporal modeling. Can the authors explain its role in the model all at once.

[1] Liu, Tianqi, et al. "Free4D: Tuning-free 4D Scene Generation with Spatial-Temporal Consistency." arXiv preprint arXiv:2503.20785 (2025).

[2] Wu, Rundi, et al. "Cat4d: Create anything in 4d with multi-view video diffusion models." Proceedings of the Computer Vision and Pattern Recognition Conference. 2025.

[3] Zhao, Yuyang, et al. "Genxd: Generating any 3d and 4d scenes." arXiv preprint arXiv:2411.02319 (2024).

[4] Sun, Wenqiang, et al. "Dimensionx: Create any 3d and 4d scenes from a single image with controllable video diffusion." arXiv preprint arXiv:2411.04928 (2024).

**Questions:**

- Lack of visualizations in supplement video. In the main paper, the author compared Recamaster and TrajectoryCrafter, but did not compare them in the supplementary video. I am curious about their dynamic effects.
- Lack of depth visualizations before and after depth refinement.

---

### Official Review · Reviewer_K7RQ · 2025-10-31

**Soundness:** 2
**Presentation:** 3
**Contribution:** 2
**Rating:** 2
**Confidence:** 4

**Summary:**

This paper presents WorldCrafter, a method that generates dynamic scene videos from a single image and enables controllable viewpoint changes through additional input camera parameters. WorldCrafter proposes a Video Depth Refinement method and represents the scene using Gaussians to ensure that the generated videos contain accurate geometric information.

**Strengths:**

- The paper is clearly written and easy to follow, with well-designed figures and tables that facilitate understanding of the proposed method.
- The paper focuses on geometric representation and temporal consistency in video generation, which are valuable research directions and key challenges that need to be addressed in current video generation and world model studies.

**Weaknesses:**

- Concerns on Method
1. The median filtering formula in lines 257–258 appears to refine the depth values within the same semantic region to a single value. Such an operation seems to severely damage the geometric structure of the scene. For example, a river region should naturally have a wide depth range, but after this depth refinement operation, it would be collapsed into a single fixed depth value. This is clearly incorrect and may significantly affect the subsequent process of lifting 2D images into 3D representations. The authors are encouraged to provide further explanation of this operation. In addition, it is suggested to include visualizations of the depth maps before and after median filtering to help readers better understand the proposed depth refinement process.
2. The paper emphasizes that the proposed method models temporal consistency. However, the described components—depth refinement, 3DGS modeling, and image inpainting—are all applied to individual frames, without any explicit modeling of inter-frame consistency. The authors should clearly specify which part of the proposed method is responsible for modeling the temporal consistency of the generated results.
3. What is the purpose of representing the scene using 3DGS? Why not adopt a point-cloud-based representation as used in methods such as TrajectoryCrafter[1]? In addition, how are the attributes of the Gaussian primitives—particularly the opacity and covariance matrix—initialized? How is the 3DGS trained?

- Concerns about Experiment
4. From the visual comparison results in the supplementary video, the outputs of WonderJourney[2] and WonderWorld[3] exhibit highly abnormal content flickering, which does not appear in the official demos of these methods. The authors should clarify why such artifacts occur and explicitly specify the implementation details and settings used for these comparison methods.
5. Although the proposed method takes a single image as input, it generates videos using an I2V model. Therefore, it is necessary to compare it with V2V methods such as ReCamMaster[4]. The paper should provide more visual comparison results with ReCamMaster. In addition, the authors should include a baseline where TrajectoryCrafter’s depth maps are replaced with the refined depth maps from this work, in order to demonstrate the effectiveness of the depth refinement and the advantages of representing the scene with 3DGS compared to point-cloud-based representations.

[1] TrajectoryCrafter: Redirecting Camera Trajectory for Monocular Videos via Diffusion Models. ICCV 2025

[2] Wonderjourney: Going from anywhere to everywhere. CVPR 2024.

[3] Wonderworld: Interactive 3d scene generation from a single image. CVPR 2025

[4] Recammaster: Camera-controlled generative rendering from a single video. ICCV 2025

**Questions:**

Please see the weekness.

---

### Official Review · Reviewer_mZU9 · 2025-11-01

**Soundness:** 2
**Presentation:** 2
**Contribution:** 2
**Rating:** 4
**Confidence:** 4

**Summary:**

This paper proposes a method to enhance the multi-view and temporal consistency in the video generation.  To this end,  the method design a reconstruct-then-synthesis approach: given an generated video conditioned on an image, it first refines its depth estimation to obtain a 3D Gaussian representation of the background scene and objects.  Then, the 3D representations can be rendered in different viewpoints, and then the rendering results are fed into the video generation model for inpainting.

**Strengths:**

1.  This paper tackles a challenging problem: removing the structure flickering in the generated video under large viewpoint change.
2.  The proposed method combines the video depth estimation,  3D Gaussian representation to reconstruct 3D structure from the initial video after removing the noise and outliers. This reconstructed 3D structure can help to increase the multi-view and temporal consistency  of generated videos.

**Weaknesses:**

1.  While the experimental results are impressive,  the advantage of this method over STOA camera trajectory control method for video generation are not comprehensive.  I would like to see more comparisons with video generation methods,  for instance,  Animateanything: Consistent and Controllable Animation for Video Generation CVPR 2025.

2. The proposed method is a post-processing method for video generation that heavily relies on 3D reconstruction. Therefore, the generalizability of this method relies on the generalizability of video depth estimation and so on.   More tests on various kind of scenes  are necessary to verify the claimed contribution in this introduction, since it is difficult to obtain accurate and consistent depth for dynamic objects with some materials, such as glass and highly  specular materials.

**Questions:**

How is the quality of generated videos related to the video depth? Besides, please also report the timing statistics of the proposed method.

---

### Official Review · Reviewer_ccvo · 2025-11-01

**Soundness:** 2
**Presentation:** 2
**Contribution:** 2
**Rating:** 2
**Confidence:** 4

**Summary:**

This paper introduces WorldCrafter, a framework for generating interactive dynamic video scenes from text prompts. The method combines a CogVideoX-based base generator, SAM-guided segmentation, DepthCrafter-based geometry estimation, and dynamic Gaussian splatting, followed by a rendering-guided inpainting stage. The goal is to enhance structural consistency, temporal coherence, and interactivity in generated videos.

**Strengths:**

The framework is well-motivated and integrates geometry-aware depth refinement and object-consistent modeling in a coherent way to improve performance on dynamic scenes. However, there are many concerns regarding the results.

**Weaknesses:**

1.	Despite geometry-aware modeling, generated videos still suffer from noticeable flickering and structural artifacts, especially around dynamic or thin regions, undermining temporal consistency claims.
2.	Most examples involve static or weakly dynamic scenes; only one example shows moderate motion. The method’s performance on complex dynamic scenarios remains unproven.
3.	Outputs are significantly blurrier than CogVideoX, lacking sharpness and high-frequency details. See the low metric number of image quality in Table 2, much worse than CogVideoX.
4.	Module-wise improvements are marginal in Table 3, raising concerns about the necessity and actual impact of each proposed component.
5.	While built on CogVideoX, the paper omits a direct comparison with its original output, making it unclear whether the overall fidelity or diversity improves.
6.  Given the artifacts present in DepthCrafter, it is unclear why more powerful prior models, such as VideoDepthAnything, were not used instead.

**Questions:**

See the weakness part.

---

### Note · Authors · 2025-11-12

I have read and agree with the venue's withdrawal policy on behalf of myself and my co-authors.